# An essential role for α4A-tubulin in platelet biogenesis

Catherine Strassel[1], Maria M Magiera[2,3] , Arnaud Dupuis[1] , Morgane Batzenschlager[1] , Agnès Hovasse[4], Irina Pleines[5,6], Paul Guéguen[7], Anita Eckly[1], Sylvie Moog[1], Léa Mallo[1], Quentin Kimmerlin[1], Stéphane Chappaz[5,6], Jean-Marc Strub[4], Natarajan Kathiresan[2,3], Henri de la Salle[1], Alain Van Dorsselaer[4], Claude Ferec[7], Jean-Yves Py[8] , Christian Gachet[1], Christine Schaeffer-Reiss[4], Benjamin T Kile[5,6], Carsten Janke[2,3] , François Lanza[1]

During platelet biogenesis, microtubules (MTs) are arranged into submembranous structures (the marginal band) that encircle the cell in a single plane. This unique MT array has no equivalent in any other mammalian cell, and the mechanisms responsible for this particular mode of assembly are not fully understood. One possibility is that platelet MTs are composed of a particular set of tubulin isotypes that carry specific posttranslational modifications. Although β1-tubulin is known to be essential, no equivalent roles of α-tubulin isotypes in platelet formation or function have so far been reported. Here, we identify α4A-tubulin as a predominant α-tubulin isotype in platelets. Similar to β1-tubulin, α4A-tubulin expression is up-regulated during the late stages of megakaryocyte differentiation. Missense mutations in the α4A-tubulin gene cause macrothrombocytopenia in mice and humans. Defects in α4A-tubulin lead to changes in tubulin tyrosination status of the platelet tubulin pool. Ultrastructural defects include reduced numbers and misarranged MT coils in the platelet marginal band. We further observed defects in megakaryocyte maturation and proplatelet formation in *Tuba4a*-mutant mice. We have, thus, discovered an α-tubulin isotype with specific and essential roles in platelet biogenesis.

## Introduction

Blood platelets are produced by megakaryocytes (MKs) in the bone marrow and are essential to arrest bleeding in mammals. In the final stages of platelet biogenesis, microtubules (MTs) are arranged into a submembranous structure, the marginal band, which encircles the platelets in a single plane. This circular scaffold is essential for maintaining the flattened lenticular shape of platelets

(1, 2) and has no equivalent in any other mammalian cell. The biological importance of the marginal band has been demonstrated by the observation that MT depolymerization with colchicine, or by cold exposure, induces a transition of platelets to a spherical shape and alters platelet function (3). In addition, decreased platelet counts and enlarged spheroid platelets have been observed in patients and animals with genetic deficiencies in the platelet-specific β1-tubulin isotype (4, 5).

MTs in platelets undergo dramatic bending when forming the marginal band; however, the underlying mechanisms remain to be elucidated. When MKs are differentiated in culture, MT coiling in the extending pseudopods occurs during proplatelet formation. Pseudopods show platelet-sized swellings at their extremities, which will later bud off as platelets (6). The dynamics of MT growth and sliding in proplatelets depend on the dynein/dynactin motor complex (7, 8), but it is unclear how MT ring formation and maintenance is regulated.

The particular mode of MT assembly in platelets suggests that tubulin building blocks might have unique biochemical and structural properties. In humans, nine α-tubulin and nine β-tubulin genes encode a variety of tubulin isotypes (9). Within the α- and the β-tubulin families, isotypes show more than 90% amino acid identity, with the notable exception of β1-tubulin. The gene encoding this particular isotype, *TUBB1*, was initially cloned from a bone marrow cDNA library (10) and subsequently demonstrated to be specifically expressed in MKs and platelets (11). It is now clear that incorporation of β1-tubulin into platelet MTs is critical for the formation of the marginal band and, therefore, for platelet formation and function. It is one of the most striking examples of a functional specialized, unique, and essential role for a tubulin isotype. Other isotypes have also been found to be specifically incorporated into selected MTs, such as β3-tubulin in neurons (12, 13), or β4-tubulin in cilia and flagella (14, 15).

[1]Université de Strasbourg, Institut National de la Santé et de la Recherche Médicale, Etablissement Français du Sang Grand Est, Unité Mixte de Recherche-S 1255, Fédération de Médecine Translationnelle de Strasbourg, Strasbourg, France   [2]Institut Curie, Paris-Sciences-et-Lettres Research University, CNRS UMR3348, Orsay, France   [3]Université Paris Sud, Université Paris-Saclay, CNRS UMR3348, Orsay, France   [4]Laboratoire de Spectrométrie de Masse BioOrganique, Institut Pluridisciplinaire Hubert Curien, CNRS UMR7178, Université de Strasbourg, Strasbourg, France   [5]ACRF Australian Cancer Research Foundation Chemical Biology Division, the Walter and Eliza Hall Institute of Medical Research, Parkville, Australia   [6]Anatomy and Developmental Biology, Monash Biomedicine Discovery Institute, Monash University, Melbourne, Australia   [7]Laboratoire de génétique moléculaire et d'histocompatibilité, Centre Hospitalier Régional et Universitaire Morvan, INSERM U1078, EFS Bretagne, Brest, France   [8]EFS Centre-Pays de la Loire, site d'Orléans, France

Correspondence: francois.lanza@efs.sante.fr; Carsten.Janke@curie.fr
Natarajan Kathiresan's present address is Physiology and Biomedical Engineering, Mayo Clinic, Rochester, MN, USA

The mechanical properties of MTs are thought to be predominantly governed by lattice interactions (16). As such, it might be expected that they could be altered by the incorporation of a particular set of tubulin isotypes into the lattice (17). Although the essential role of β1-tubulin for platelet MTs has so far only been determined by genetic approaches, it is highly likely that its incorporation changes the mechanical and dynamic properties of platelet MTs.

Strikingly, nothing is known about the role of α-tubulin isotypes in platelets and MKs. The reason for the lack of insight into the α-tubulin repertoire is most likely related to the fact that so far, no reliable methods are available to quantitatively characterize the composition of the tubulin isotype pool, and that DNA and peptide sequences of α-tubulin isotypes are even more similar among each other than those of β-tubulins, making quantitative analyses challenging. To detect isotypes that could be important for platelet biogenesis, we investigated the composition of the tubulin isotype pool of platelet MTs with a specific focus on the yet unexplored α-tubulin isotypes. Using quantitative proteomics, we found a strong enrichment of α4A-tubulin in platelets. The expression of α4A-tubulin is up-regulated during the late stages of MK differentiation, and we found an enrichment of α4A-tubulin in platelets. This is reminiscent of the profile of β1-tubulin expression, suggesting that α4A-tubulin plays essential roles in platelets.

Indeed, mice carrying a missense mutation in the *Tuba4a* gene, and a patient with a monoallelic double missense mutation in the *TUBA4A* gene displayed macrothrombocytopenia and structural abnormalities in the marginal band of platelets. In addition, defective MK maturation and proplatelet formation were observed in the mutant mouse model. The fact that these phenotypes are highly similar to the β1-tubulin–mutant phenotypes implies a similarly essential role of α4A-tubulin in late stages of platelet biogenesis.

# Materials and Methods

### Mice

*Plt68* mice carrying a mutant *Tuba4a* allele were generated. Male BALB/c mice were injected i.p. with 85 mg/kg N-ethyl-N-nitrosourea (ENU) weekly for 3 wk and rested for 12 wk before mating with untreated BALB/c females to produce first-generation progeny. Blood was taken at 7 wk of age, and animals exhibiting aberrant platelet counts and morphology were test-mated to determine heritability of the phenotype. Confirmed mutant strains were backcrossed for 10 generations to the BALB/c background to breed out irrelevant ENU-induced mutations.

### Reagents

Polyclonal antisera against mouse β1-tubulin (pAb5274) and human/mouse α4A-tubulin (pAb7620 and pAb7621) were generated by immunizing against peptides 451-VLEEDEEVTEEAEMEPEDKGH-471 and 76-DEIRNGPYR-84, respectively.

### 2D-gel electrophoresis and liquid chromatography-mass spectrometry/mass spectrometry analysis of platelet tubulin

2D-gel electrophoresis was performed according to Magnin-Robert et al (18). Spots of interest were extruded and submitted to in-gel digestion (19) and then processed for mass spectrometry as described in the Experimental Procedures section of the Supplementary Information.

### LC-Single Reaction Monitoring (SRM) quantification of α-tubulin isotypes

For LC-SRM assay, proteotypic peptide (EIIDPVLDR and DVNVAIAAIK) signatures of α4A- and α8-tubulin, respectively, and one peptide (NLDIERPTYTNLNR) common to all α-tubulins were selected. The samples were processed for LC-SRM quantification as described in the Experimental Procedures section of the Supplementary Information.

### MK culture

Human CD34[+] cells were cultured as described previously (20). Mouse bone marrow Lin[−] cells were cultured as described previously (21). Additional information is provided in the Experimental Procedures section of the Supplementary Information.

### Proplatelet formation in bone marrow explants

Marrows from femurs were flushed and cut in transverse sections of 0.5 mm. MKs were observed after 6-h incubation at 37°C, at the periphery of the tissue as described previously (22).

### Immunofluorescence microscopy

Blood was obtained from anesthetized mice and platelets were washed according to Cazenave et al (23). Platelets were fixed in 4% PFA, cytospun onto poly-L-lysine–coated slides and processed for immunofluorescence microscopy as described in the Experimental Procedures section of the Supplementary Information.

### Western blotting

Proteins, corresponding to 10[7] platelets or 300 ng of purified tubulins were separated on 4%–15% or 10% SDS gels for separation of α and β-tubulin (24), blotted onto polyvinylidene difluoride membranes, and processed for Western blotting as described in the Experimental Procedures section of the Supplementary Information.

### Screening and gene sequencing of blood donors

Blood donors with a platelet count below 150 G/L were selected, DNA extracted from buccal swabs, and 17 selected genes suspected to be implicated in thrombocytopenia, including *TUBA4A*, were amplified by PCR using amplimers encompassing the coding sequences, intron–exon junctions, and 5′ and 3′ UTR (ABC study approved by ANSM-French agency for the safety of drugs and health products registered under identification number RCB: 2014-A00002-45; informed consent was obtained for each individual).

### Statistical analyses

Results were expressed as the mean (±SEM) and statistical comparisons were performed using an unpaired, two-tailed $t$ test or a one-way ANOVA followed by the Bonferroni post hoc test.

# Results

### Analysis of the α-tubulin isotype repertoire in circulating platelets and during megakaryopoiesis

Previous studies in mouse have reported an up-regulation of β1-tubulin expression during megakaryopoiesis and estimated that this isotype represented a large proportion of total β-tubulin pool in circulating platelets (5). In contrast, a quantification of α-tubulin isotypes in platelets and MKs has so far not been performed. In a first series of experiments, we established the full repertoire of α-isotypes by nanoLC-MS/MS analysis. Tubulin purified from human platelets was separated by 2D electrophoresis as described in Adessi et al (25), and the main Coomassie-stained spots (Fig 1A) were dissected and submitted to nanoLC-MS/MS analysis. We identified α1C-, α3C/D-, α3E-, α4A-, and α8-tubulin isotypes. This represents virtually all known α-isotypes, bearing in mind that highly similar isotype groups are not distinguishable with this method.

To determine the dynamics of α-tubulin isotype expression, we quantified mRNA levels during MK differentiation of human CD34+ cells (20). Tubulin isotypes within highly homologous subgroups (*TUBA1* and *TUBA3*) were analyzed together using cross-reactive primers. The expression of isotypes from the *TUBA1* group was already detected at D0, increased early during differentiation (D4), and then remained stable until full maturation (D12; Fig 1B). In contrast, expression of *TUBA4A* and *TUBA8* transcripts progressively increased from D4 until D12, a stage just preceding proplatelet extension and marginal band formation. *TUBA3* transcripts were not detected throughout differentiation (data not shown). Strikingly, the expression profile of *TUBA4A* and *TUBA8* mirrors *TUBB1*, thus providing a further indication for the potential functional complementarity of these isotypes.

To determine the prevalence of α4A- and α8-tubulin isotypes at the protein level, we submitted tubulin purified from human platelets to a quantitation assay using a targeted mass spectrometry approach called SRM with quantified stable isotope-labeled peptides (AQUA peptides) as internal standards (26). Using specific signatures from α4A- and α8-peptides, and normalizing them to a common peptide shared by all α-tubulin isotypes, LC-SRM analysis showed a sizable fraction of α4A-tubulin (29 ± 6%, n = 6), but only low levels of α8-tubulin (2.0 ± 1.4%, n = 6) in the platelet tubulin pool (Fig 1C). This is striking as so far α4A-tubulin had been reported to be expressed in many tissues, however, generally at low transcript levels compared with other α-tubulins (27).

Using an α4A-tubulin–specific antibody that we had raised against a sequence unique to α4A-tubulin (76-DEIRNGPYR-84, pAb7621), we compared the level of α4A-tubulin in MTs purified from platelets, brain tissue, and HeLa cells. Western blot analysis

revealed that the levels of α4A-tubulin in platelet MTs are well above those in HeLa cells and also higher than those in brain (Fig 1D). This confirmed that α4A-tubulin is particularly enriched in platelet MTs and indicates that it might play a particular role in the formation of the marginal band.

### A missense mutation in *Tuba4a* gene causes macrothrombocytopenia and abnormal marginal band formation in mice

To establish the role of α4A-tubulin in the formation of the marginal band, we analyzed a mouse strain (named *Plt68*) that we had identified from an ENU mutagenesis screen. *Plt68* mice exhibited decreased platelet counts (a 21% reduction compared with wild-type mice, 943 × 10$^3$/μl versus 1,199 × 10$^3$/μl; Fig 2A). In addition, platelets from these mice exhibited an increased size with mean platelet volumes (MPV) representing 143% of those in wild-type mice (7.0 fL versus 4.9 fL; Fig 2B).

Exome sequencing of two *Plt68* animals, followed by genotyping of candidate mutations in siblings and offspring revealed that the *Plt68* phenotype correlated with inheritance of a point mutation in the *Tuba4a* gene. The mutation is predicted to encode a valine-to-glutamate substitution at position 260 of α4A-tubulin (P68366; Fig 2C).

We next examined the ultrastructure of *Tuba4a*$^{V260E/V260E}$ platelets by scanning and transmission electron microscopy. This analysis confirmed the increased size of the platelets and revealed the loss of their typical flat, discoid shape (Fig 2D). Close-up views of transmission electron microscopy (TEM) cross sections (Fig 2E) revealed a reduction in the number of MT coils in the marginal band of *Tuba4a*$^{V260E/V260E}$ platelets (average 5 ± 1.0 coils/platelet compared with 11 ± 0.6 coils in the wild-type). Similar defects have been described for mice lacking β1-tubulin (5) (Fig S1) or patients harboring mutations in the *TUBB1* gene (4). Consequently, these results imply that α4A-tubulin has, similar to β1-tubulin, an essential role in the formation of the marginal band during platelet biogenesis.

### Naturally occurring mutations of *TUBA4A* in a human individual with moderate macrothrombocytopenia

To determine whether these observations in mice are relevant to humans, we screened for *TUBA4A* mutations in individuals presenting with low platelet counts. Blood donors with subthreshold platelet counts (<150 × 10$^3$/μl) were subjected to targeted sequencing of genes possibly implicated in thrombocytopenia. One individual was identified with a monoallelic c.541G>A and c.547G>C double substitution in *TUBA4A* predicting p.Val181Met and p. Glu183Gln mutations in α4A-tubulin (Fig 3A). These mutations were confirmed by Sanger sequencing. Database searches indicated that they correspond to very rare variants, with frequencies of 8e-5 and 2e-5 for c.541G>A (rs757373635) and c.547G>C (rs775821469), respectively.

Platelet counts recorded for the patient between 2008 and 2017 (n = 26) ranged between 137 and 164 × 10$^3$/μl, but MPVs were not available. Follow-up evaluation of the blood parameters confirmed a platelet count at the lower limit of normal values (158 × 10$^3$/μl), considering normal ranges for platelet counts of 172–398 × 10$^3$/μl in a French adult male population (16–59 y: n = 17,646) (28) and

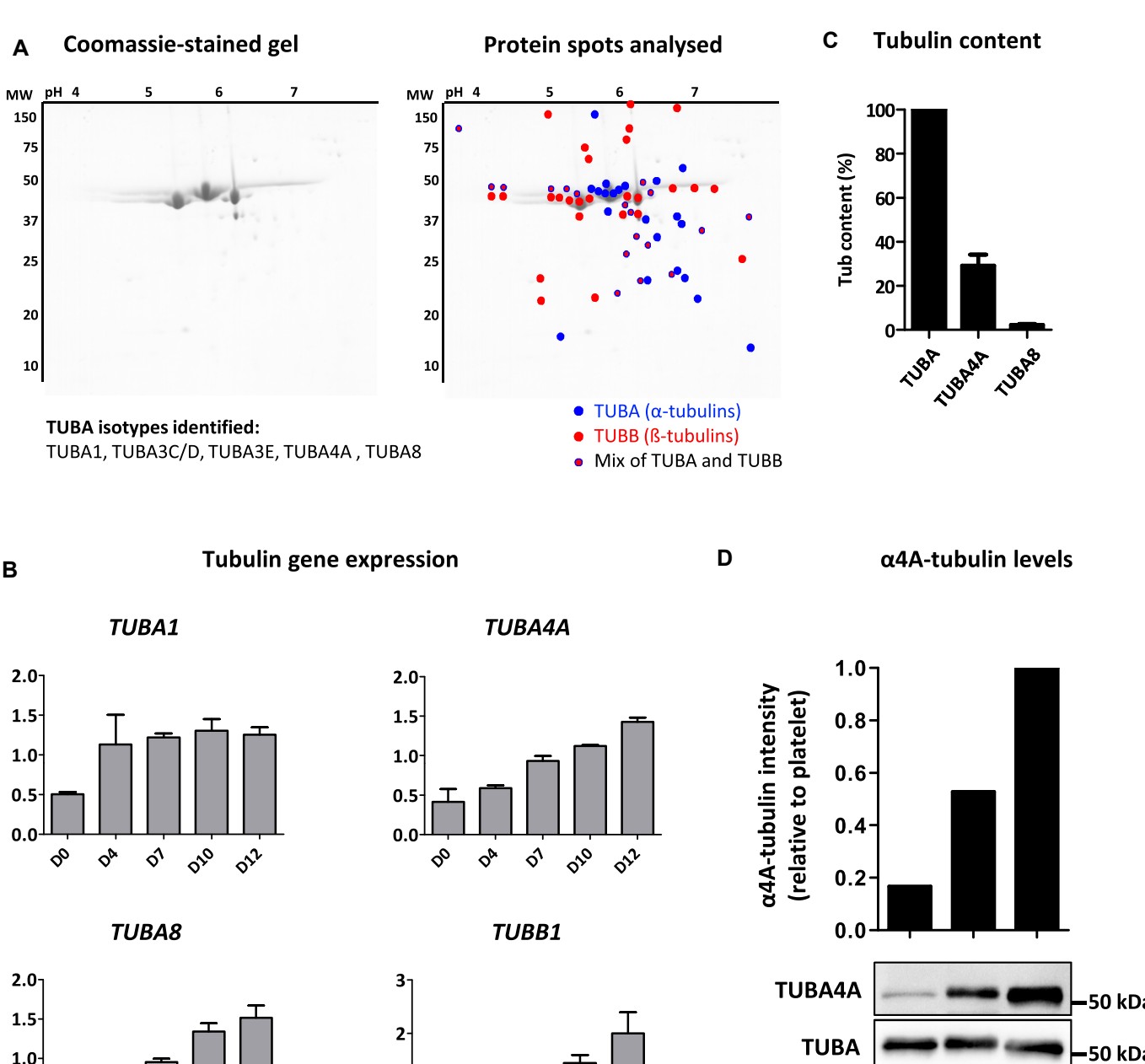

**Figure 1. Expression of α-tubulin isotypes in platelets and during megakaryopoiesis.**

**(A)** *2D-gel electrophoresis of tubulins purified from human platelets and list of identified α-isotypes.* The isotypes were identified by LC-MS/MS analysis of the color-marked spots (blue, red, and double labelling correspond to spots containing α-, β-, and αβ isotypes, respectively). **(B)** *Evolution of α1-, α4A-, α8-, and β1-tubulin transcripts at different stages of MK differentiation.* RT-PCR amplification of RNA isolated from MKs differentiated from human CD34⁺ progenitor cells at days 0 to 12 of culture. Bands were quantified on gels and intensity normalized to 18S (mean ± SEM, n = 3). **(C)** *SRM-MS quantification of the α4A- and α8-tubulin isotypes in tubulin purified from human platelets.* The content normalized to the total α-tubulin content was calculated as described in the methods from six separate tubulin preparations with analyses performed in triplicate. **(D)** *Western blot analysis of α4A-tubulin levels in tubulin purified from platelets, the brain, and HeLa cells.* (Upper panel) Quantification of α4A-tubulin levels in HeLa and brain tubulin relative to platelet tubulin after normalization to the total α-tubulin content. (Lower panel) Representative blot where equivalent amounts of purified tubulin (300 ng) were separated on a 10% gel and probed using pAb7621 polyclonal Ab specific for α4A-tubulin and DM1a mouse mAb recognizing all the α-tubulin isotypes.

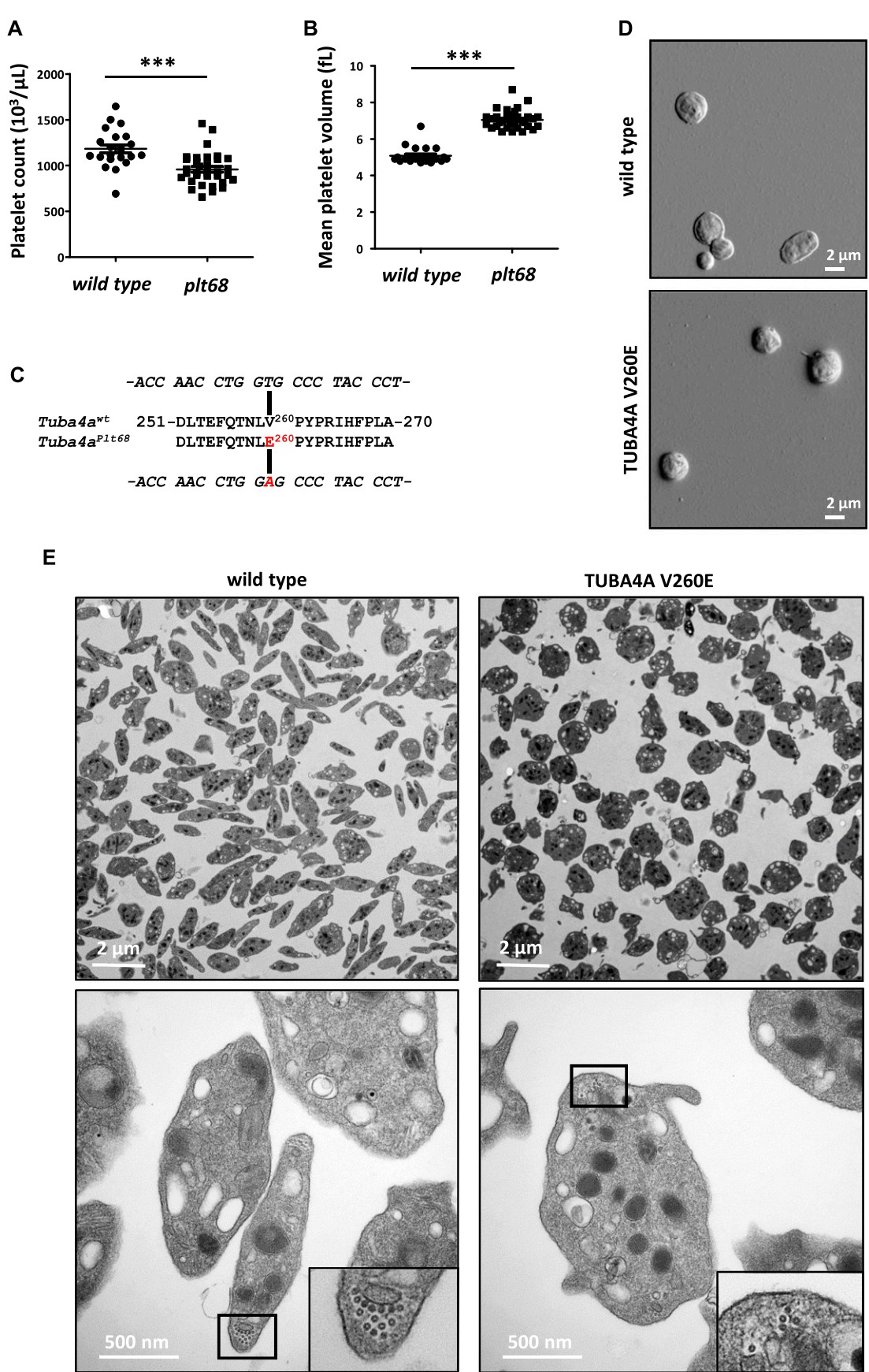

A

B

C

-ACC AAC CTG GTG CCC TAC CCT-

$Tuba4a^{wt}$  251-DLTEFQTNLV$^{260}$PYPRIHFPLA-270
$Tuba4a^{plt68}$     DLTEFQTNL**E**$^{260}$PYPRIHFPLA

-ACC AAC CTG G**A**G CCC TAC CCT-

D

wild type

TUBA4A V260E

E

wild type

TUBA4A V260E

revealed an increased mean platelet volume (12.9 fL; normal range: 7.8–10.8 fL) (28). Electron microscopy analyses of the patient's platelets revealed similar defects to those in *Tuba4a*[V260E/V260E] mice with a loss of the typical flat, discoid shape (Fig 3B and C). TEM cross sections did not, however, reveal a decrease in the number of MT per platelet (10.9 ± 0.4 for patient versus 11.4 ± 0.5 for control), but rather a profound disorganization of the marginal band, featuring loosely assembled MTs (Fig 3C). Together with the results of our mouse model, these findings strongly support an evolutionarily conserved key role of α4A-tubulin in platelet biogenesis.

### Effect of *Tuba4a* mutations on α4A-tubulin expression and on tyrosination of the α-tubulin pool

We next evaluated the consequences of *Tuba4a*[V260E] and *TUBA4A* mutations on the expression of α4A-tubulin by Western blotting using isotype-specific antibodies. Although we detected a specific 55-kD protein band in platelet extracts from wild-type mice (Fig 4A) and normal human individuals (Fig 4B), this band was not detected in platelet extracts from *Tuba4a*[V260E/V260E] mice, and the signal was decreased in extracts from the patient. qRT-PCR performed in cultured MK did not reveal a decrease in Tuba4a transcript levels in *Tuba4a*[V260E] mice and additionally confirmed the increase of Tuba4a during differentiation (Fig S2), as in human cultured MK. Consistent with this, a dramatic reduction in labelling of the marginal band using the α4A-tubulin antibodies was observed in mutant platelets relative to those from wild-type mice (Fig 4C) or human control individuals (Fig 4D). Other α-tubulin isotypes were present at relatively normal levels in extracts from *Tuba4a*[V260E/V260E] mice and the patient, as seen with the pan-α-tubulin DM1A antibody (29). The fact that overall α-tubulin levels are unaltered in the absence of TubA4A is most likely related to the adaptation of the expression levels of the remaining α-tubulin isotypes (30, 31). We also determined the expression of β1-tubulin and found no obvious changes.

The nearly complete absence of α4A-tubulin in *Tuba4a*[V260E/V260E] mice could be caused by the misfolding of the protein and subsequent degradation. We thus performed 3D modelling of Tuba4a[V260E/V260E], which shows that the V260E mutation is situated within the core region of the protein (Fig S3). Thus, the substitution of hydrophobic valine by a charged glutamate might destabilize the entire core region, thereby affecting the conserved tubulin fold, leading to misfolded tubulin being degraded (32).

Compared with other α-tubulins, α4A-tubulin shows a rather high level of sequence conservation. However, an outstanding feature of α4A-tubulin is the lack of a C-terminal tyrosine residue, which is present on all other α-tubulin isotypes (or substituted by a functionally similar phenylalanine). Tyrosinated α-tubulin isotypes can undergo enzymatic posttranslational detyrosination and retyrosination, and strict regulation of this "tyrosination cycle" is essential for a variety of MT functions (reviewed in reference 33). A strong expression of α4A-tubulin might significantly increase the initial levels of detyrosinated tubulin in MKs and platelets, thus affecting tyrosination-dependent MT functions. We examined the levels of tyrosinated and detyrosinated tubulin in platelets of *Tuba4a*[V260E/V260E] mice and from the patient. Western blotting analysis with an antibody directed against tyrosinated α-tubulin (Tyr-tubulin) showed stronger labelling in both cases relative to controls which implies a significant increase in tyrosination of the α-tubulin pool (Fig 4A and B). Conversely, the level of detyrosinated tubulin was decreased in *Tuba4a*[V260E/V260E] platelets, but less so in the patient's platelets. It thus appears that the high proportion of detyrosinated α-tubulin in wild-type platelets might primarily be the result of the strong expression of α4A-tubulin in platelets, and absence or reduction of this isotype strongly affects the levels of this otherwise posttranslationally controlled modification of tubulin.

Acetylation is another tubulin posttranslational modification present in native resting platelets (34). It has been proposed to play a regulatory role during platelet activation and shape change (35), whereas its role in platelet biogenesis is still debated (36, 37). We verified the levels of tubulin acetylation between wild-type and *Tuba4a*[V260E/V260E] platelets but could not detect significant differences (Fig S4). This suggests that tubulin acetylation is not involved in the defects observed in *Tuba4a*[V260E/V260E] mice.

### α4A-tubulin–deficient MKs have a reduced capacity to extend proplatelets

To determine the impact of α4A-tubulin deficiency on the process of platelet formation, we analyzed each step of megakaryopoiesis in *Tuba4a*[V260E/V260E] mice. The bone marrow showed a normal cellular distribution of morphologically distinguishable MKs (stages I–III; Fig 5A), and no ultrastructural defects were noted in the less mature MKs (stages I–II). In contrast, a notable proportion (~30%) of stage III MKs exhibited an abnormal structure with a more compact demarcation membrane system (Fig 5B). When we examined the capacity of bone marrow–derived MKs to extend proplatelets by real-time analysis of fresh bone marrow explants (22), we noted a profound defect in *Tuba4a*[V260E/V260E] MKs: MKs from these mice were unable to extend well-developed proplatelets, which are typically observed in wild-type MKs at this stage (Fig 6A). Most of the *Tuba4a*[V260E/V260E] MKs retained a spherical shape during the 360-min course of observation, and only a small proportion extended a few thick protrusions. This phenotype is characteristic for defective proplatelet formation.

**Figure 2. An ENU-induced mutation of *Tuba4a* causes macrothrombocytopenia and abnormal marginal band formation.**
**(A)** *Decreased platelet counts in Plt68 mice.* The number of circulating platelets in wild-type and *Plt68* mice is represented for each individual mouse (1,199 ± 183 versus 943 ± 207 × 10³ platelets/μl, respectively; mean ± SEM n = 21 wild type and n = 33 *Plt68*; ***P = 0.0026; t test). **(B)** *Increased platelet volume in Plt68 mice.* The mean platelet volume in wild-type and *Plt68* mice is represented for each individual mouse (4.9 ± 0.34 versus 7.0 ± 0.17 fL respectively; mean ± SEM n = 21 and n = 33; ***P < 0.0001; t test). **(C)** *Schematic representation of the V260E mutation:* A mutation was identified in the gene encoding α4A-tubulin (*Tuba4a*) in the *Plt68* strain resulting in a Val-to-Glu transition at position 260 of the protein (P68366). **(D)** Representative scanning electron microscopy images of wild-type and *Tuba4a*[V260E/V260E] platelets. **(E)** Representative transmission electron microscopy images of wild-type and *Tuba4a*[V260E/V260E] platelet suspensions (upper panels) and close-up views of individual platelets in cross section (lower panels).

**A**

```
              -TCT ACA GCC GTG GTC GAG CCC TAC AAC
                                \       \
TUBA4A   171-IYPAPQVSTA V¹⁸¹V E¹⁸³ PYNSILT-190
patient      IYPAPQVSTA M¹⁸¹V Q¹⁸³ PYNSILT
                                /       /
              -TCT ACA GCC ATG GTC CAG CCC TAC AAC
```

**B**

control | patient

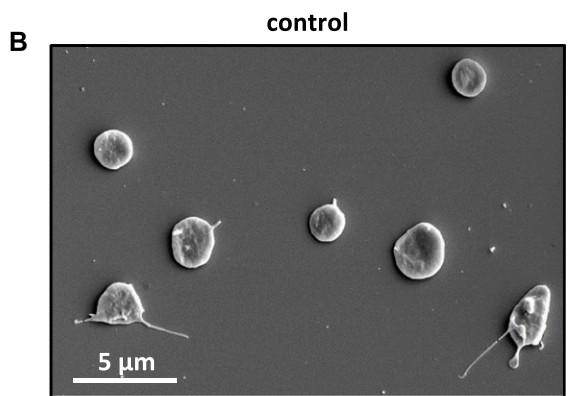 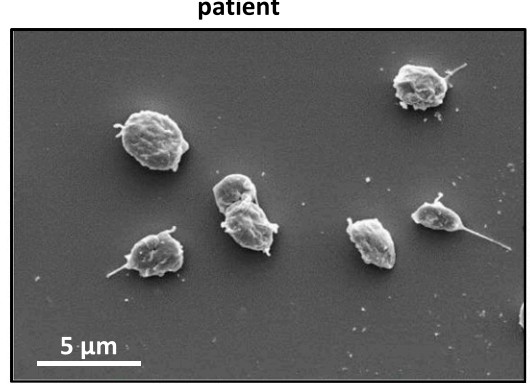

5 μm | 5 μm

**C**

control | patient

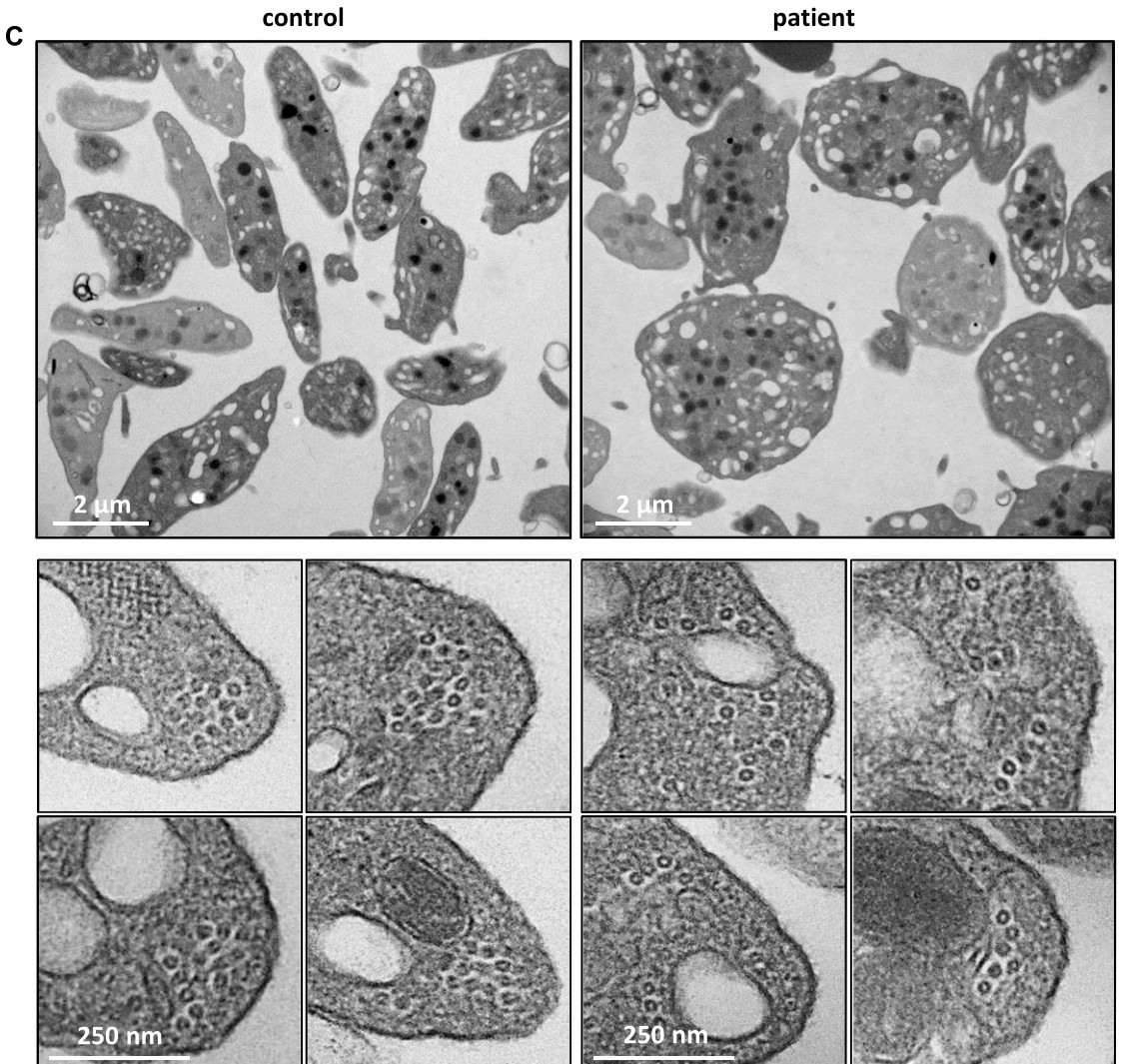

2 μm | 2 μm

250 nm | 250 nm

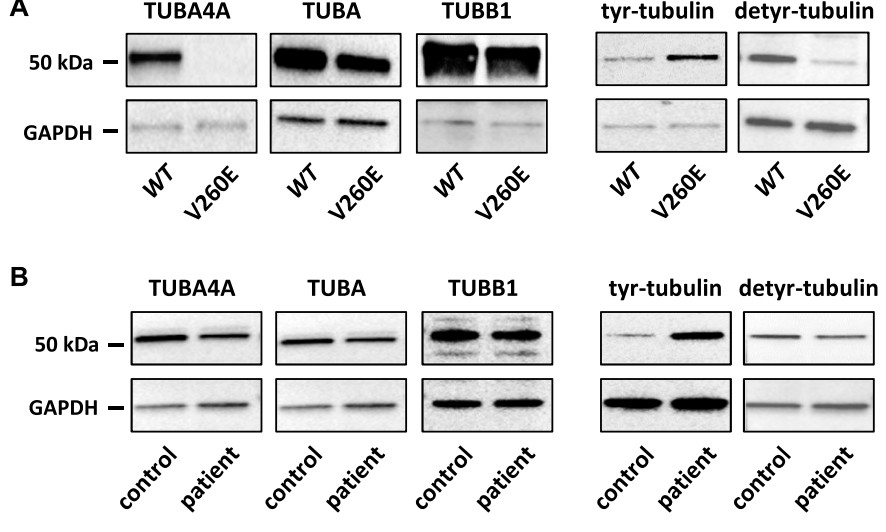

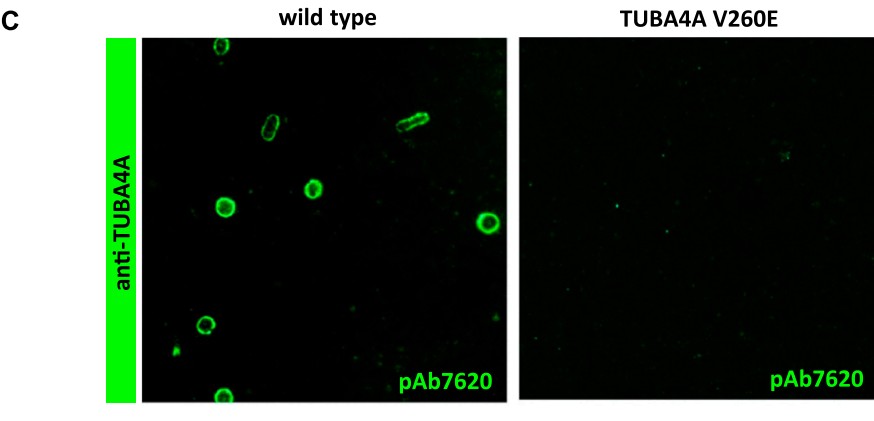

**Figure 4. Effect of Tuba4a mutations on α4A-tubulin expression and on tyrosination of the α-tubulin pool.**
**(A)** *α4A-tubulin is not detected in platelets from Tuba4a*$^{V260E/V260E}$ *mice and the α-tubulin pool is hypertyrosinated.* Platelet lysates from wild-type and *Tuba4a*$^{V260E/V260E}$ mice were separated by SDS-PAGE and probed with pAb7621 against α4A-tubulin, DM1a recognizing all the α-tubulin isotypes, pAb5274 against β1-tubulin, 1.A2 against tyrosinated α-tubulin, or 1D5 against detyrosinated α-tubulin. Blots were also probed for GAPDH as a loading control. Representative of four separate experiments. **(B)** *α4A-tubulin is decreased in platelets from the patient and the α-tubulin pool is hypertyrosinated.* Platelet lysates from a control individual and the patient were processed as in (A). **(C, D)** *Decreased α4A-tubulin labelling in the marginal band of platelets from Tuba4a*$^{V260E/V260E}$ *mice (C) and the patient (D).* Platelets were fixed in PFA-Triton X-100, captured on poly-L lysine–coated slides, incubated with pAb7620 or pAb7621 against α4A-tubulin and revealed with GAR-Alexa 488.

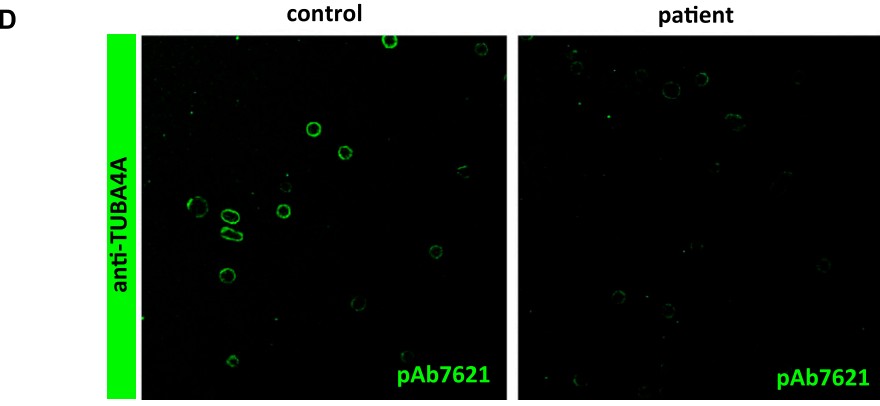

We next examined the impact of α4A-tubulin deficiency in MKs differentiated from Lin⁻ bone marrow progenitors in vitro (21). Strikingly, a defect similar to that observed in explants was observed after 4 d in culture, a time point at which proplatelet formation takes place (Fig 6B). Almost 40% of *Tuba4a*$^{V260E/V260E}$ MKs extended only few thick protrusions, indicative of poorly

**Figure 3. Naturally occurring mutations of *TUBA4A* in an individual with mild macrothrombocytopenia.**
**(A)** A double substitution was identified in *TUBA4A* in the patient, resulting in p.Val181Met and p.Glu183Gln changes in α4A-tubulin. **(B)** Representative scanning electron microscopy images of control and patient's platelets. **(C)** Representative transmission electron microscopy images of control and patient's platelet suspensions (upper panels) and close-up views of individual platelets in cross section (lower panels).

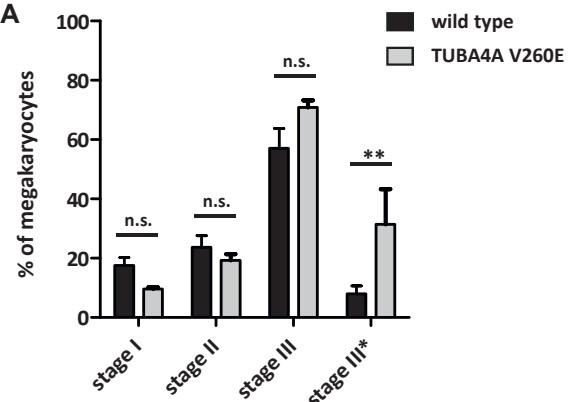

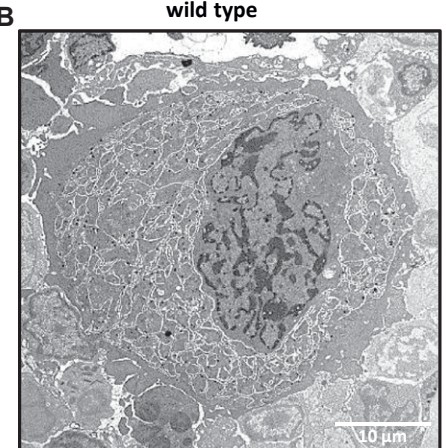
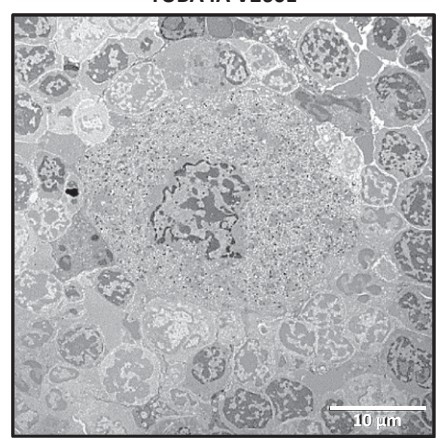

**Figure 5. Abnormal ultrastructure of stage III bone marrow MKs in *Tuba4a^{V260E/V260E}* mice.**
**(A)** *Distribution of bone marrow MKs according to their stage of differentiation (I–III).* MKs were staged in wild-type and *Tuba4a^{V260E/V260E}* mice according to morphological criteria (described in the Materials and Methods section) from transmission electron microscopy examination of bone marrow sections. Data expressed as the percentage of each stage correspond to 80 and 99 cells analyzed, respectively. Stage III* corresponds to MKs with an abnormal ultrastructure as represented in panel B. (**P < 0.001; two-way ANOVA with Bonferroni posttest). **(B)** *Representative transmission electron microscopy images of stage III MKs.* A proportion of stage III MKs exhibited abnormal ultrastructural features in *Tuba4a^{V260E/V260E}* mice as compared with wild-type mice, characterized by a smaller size, a more condensed/compact demarcation membrane system, and lack of peripheral zone.

developed proplatelets, whereas this phenotype was observed in less than 10% of wild-type MKs. This confirmed the notion that in the absence of α4A-tubulin, the early differentiation program of MKs is maintained, whereas the later stages of MK maturation and proplatelet formation are affected. The functional importance of α4A-tubulin in later stages of MK maturation coincides with the increased expression of *TUBA4A* in these developmental stages (Fig 1B).

## Discussion

Our knowledge of the distribution and function of tubulin isotypes is very rudimentary. In platelets, β1-tubulin was found to be the major β-tubulin isotype, and its expression was, for a long time, thought to be restricted to the MK lineage (11). Recently, *TUBB1* has also been detected in the thyroid, where it is important for the development of the thyroid gland (38). Studies of β1-tubulin–deficient mice, as well as the identification of patients carrying mutation in the *TUBB1* gene provided strong support for its functional importance in platelets (4, 5). In contrast, nothing was so far known about the role of specific α-tubulin isotypes, which are more difficult to quantify because of the extraordinary conservation of their peptide sequences. We overcame this obstacle by developing a quantitative mass spectrometry assay, revealing a

strong α4A-tubulin enrichment in platelets. So far, *TUBA4A* had been found most abundantly only in the heart and muscle (27). Our data thus reveal a unique isotype composition of platelet tubulin, with high proportions of α4A- and β1-tubulin that could account for the unique structural and dynamic properties of MTs in their marginal band. However, in contrast to β1-tubulin, which shows a strong sequence divergence from all other β-tubulin isotypes, the amino acid sequence of α4A-tubulin is similar to most other α-tubulin isotypes. It is, thus, puzzling why its absence cannot be compensated by other α-tubulins.

There is, however, one unique feature distinguishing α4A-tubulin from all other α-tubulin isotypes: it does not encode the ultimate C-terminal tyrosine residue. This implies that the prominent expression of α4A-tubulin in platelets shifts the tubulin pool toward detyrosinated α-tubulin, which we indeed demonstrate in α4A-tubulin–deficient platelets in both, the mutant mouse and the patient. Perturbing the tyrosination/detyrosination balance can have dramatic impacts: mice lacking tubulin tyrosine ligase show strong accumulation of detyrosinated α-tubulin and die postnatally. Neurons from these mice show uncontrolled and premature neurite outgrowth due to perturbed MT functions (39). Considering that the degree of tyrosination modulates bending flexibility of MTs in muscles (40), it is conceivable that the expression of *TUBA4A* regulates the bending of MTs during platelet formation.

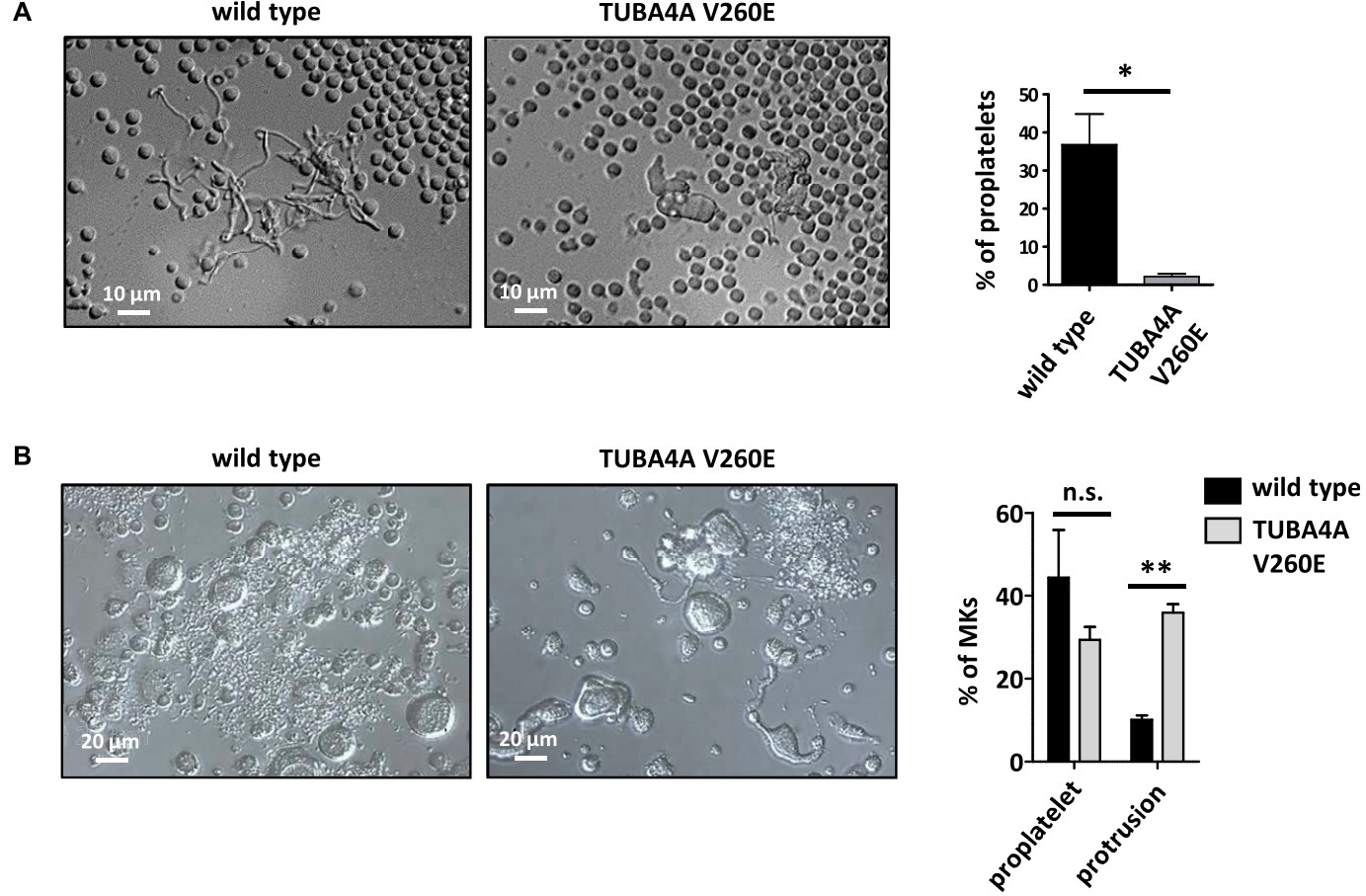

**Figure 6. Impaired proplatelet formation in *Tuba4a^V260E/V260E* mice.**
**(A)** *Proplatelet extensions in bone marrow explants.* (left) Representative DIC microscopy images of MKs observed at the edge of a bone marrow slice showing well-developed proplatelets in the wild-type and lack of pseudopodial extensions in *Tuba4a^V260E/V260E* mice. (right) The graph represents the percentage of MKs displaying proplatelets (wild type: 36.8 ± 8.1% and *Tuba4a^V260E/V260E*: 2.1 ± 0.7%; N = 494 and 495 MKs, respectively; *P = 0.0112, t test). **(B)** *Proplatelet extensions of MKs cultured from Lin⁻ progenitors.* (left) Representative DIC microscopy images of MKs cultured for 4 d from Lin⁻ progenitors showing MKs with abnormal shapes, displaying thick protrusions, in *Tuba4a^V260E/V260E* mice instead of well-developed proplatelets as seen in the WT. (right) The graph represents the percentage of MKs displaying proplatelets and thick protrusions (WT: 44.3 ± 11.5% and 10.1 ± 1.1%, respectively; *Tuba4a^V260E/V260E*: 29.3 ± 3.1% and 35.9 ± 2.1%, respectively; values from 213 WT and 162 *Tuba4a^V260E/V260E* MKs). (**P < 0.001; two-way ANOVA with Bonferroni posttest).

A comparison of *Tuba4a^V260E/V260E* and *Tubb1*-KO mice revealed striking similarities in platelet and MK defects. In both mouse models, platelets showed round morphology and decreased numbers of MT coils in marginal bands. The *TUBA4A* mutations in the patient also had a strong impact on platelet size and morphology but did not result in a lower number of MT coils. Similar observations were also made in several patients with *TUBB1* mutations (38), thus confirming the similarity of the defects caused by dysfunctions of these two tubulin isotypes. As mechanisms controlling the number of coils are still unknown, it is hard to predict how the absence or reduction of α4A-tubulin could induce this phenotype.

The increased size of platelets in both *Tuba4a^V260E/V260E* (29.6% diameter increase) and *Tubb1*-KO (+30.4% diameter increase; Fig S1) suggests that both tubulin isotypes are essential to perform the mechanical task of bending the MTs sufficiently to obtain the correct platelet size and MT coil number. How α4A- and β1-tubulin participate in this process could be different: β1-tubulin with its

highly divergent peptide sequence could directly change the biophysical parameters of the MT lattice. α4A-tubulin, in contrast, is highly similar to other α-tubulin isotypes; thus, the lack of the C-terminal tyrosine might be essential for its function.

Mice and individuals defective in α4A- (this study) and β1-tubulin (4, 5) similarly exhibit decreased platelet counts of moderate severity, which varies depending on the particular mutation. They can range from 45% in adult *Tubb1*-KO mice (Fig S1B) to subnormal values in the patient reported here, and in some previously described patients carrying *TUBB1* mutations (38). Inefficient proplatelet formation as shown in cultured MK from *Tuba4a^V260E/V260E*, *Tubb1*-KO, and *TUBB1* patients could contribute to lower platelet release into the circulation. Surprisingly though, the platelet counts in blood are typically less affected as it could be expected from the important defects in proplatelets observed in cultured MKs. Indeed, discrepancies between effects observed in MK culture and in vivo have been documented for other mouse models with platelet defects (41). Although in vitro models might thus not fully

reproduce the in vivo situation, they can reveal subtle defects that are not easily detected as pathologies in the first place, but which can become damaging in longer terms, that is, in ageing, or in stress related to other pathologies. Altogether, similarities between mouse models and patients with β1- and α4A-tubulin deficiencies strongly indicate that both isotypes are essential for the correct formation of the marginal band during the final stages of MK maturation. This implies that the isotype composition of both, α- and β-tubulin pool, is essential for correct platelet biogenesis.

We thus showed that macrothrombocytopenia is a common salient feature of β1-tubulin (4, 5) and α4A-tubulin deficiencies. Hereditary macrothrombocytopenia represent a group of rare diseases for which a number of gene abnormalities have so far been identified, including genes encoding transcription factors, receptors and cytoskeletal proteins (42). However, genetic causes for defects in platelet numbers and size still remain uncharacterized for approximately 50% of the patients (43). Considering that this is the first report of a *TUBA4A* mutation in a single patient, these data will need independent validation from different patients to more firmly link the mouse model to the human pathology. To this end, genetic studies targeting this gene will be required to identify additional cases of hereditary macrothrombocytopenia of yet unknown origin caused by defects in *TUBA4*. How α4A- and β1-tubulin control the structure and biophysical properties of MTs in platelets remains to be determined. Considering that α4A-tubulin affects the levels of tubulin detyrosination, our findings underpin the importance of the tubulin code in the functional regulation of MTs and emphasize the key role of this code for proper platelet function.

## Supplementary Information

## Acknowledgements

This work was supported by funds from the Association de Recherche et Développement en Médecine et Santé Publique (ARMESA to M Batzenschlager), has received support under the program Investissements d'Avenir launched by the French government and implemented by the French National Research Agency (ANR) with the references ANR-10-LBX-0038 and ANR-10-IDEX-0001-02 PSL, and was supported by a Program Grant (1113577) and Fellowship (1063008) from the Australian National Health and Medical Research Council and the DHB Foundation managed by ANZ Trustees. The work of C Janke was supported by the Institut Curie, ANR award ANR-12-BSV2-0007, and the Institut National du Cancer grants 2013-1-PL BIO-02-ICR-1 and 2014-PL BIO-11-ICR-1. We also thank the Proteomic French Infrastructure (ANR-10-INSB-08-03) for financial support. We thank Monique Freund for managing the animal facility at the EFS; Catherine Ziessel, Ketty Knez-Hippert, Jean-Yves Rinkel, Fabienne Proamer, and Véronique Heim for technical assistance; and Catherine Léon and Pierre Mangin for careful reading of the manuscript. We express our gratitude to Ramesh A Shivdasani for providing the *Tubb1* KO strain. We thank Valérie Proulle, Frédéric Adam, and Christelle Reperant for their help in preparing platelets from the patient and to Aurore Desprès for the gene sequencing experiments.

## Author Contributions

C Strassel: supervision, methodology, and writing—review and editing.
MM Magiera: investigation and methodology.
A Dupuis: investigation.
M Batzenschlager: investigation.
A Hovasse: investigation.
I Pleines: investigation.
P Guéguen: data curation, supervision, and investigation.
A Eckly: investigation and methodology.
S Moog: methodology.
L Mallo: methodology.
Q Kimmerlin: investigation.
S Chappaz: investigation.
J-M Strub: investigation.
N Kathiresan: investigation.
H de la Salle: data curation.
A Van Dorsselaer: supervision.
C Ferec: conceptualization and supervision.
J-Y Py: conceptualization.
C Gachet: supervision and writing—review and editing.
C Schaeffer-Reiss: conceptualization, supervision, and writing—review and editing.
BT Kile: conceptualization, supervision, and writing—review and editing.
C Janke: conceptualization, supervision, validation, and writing—original draft, review, and editing.
F Lanza: conceptualization, data curation, supervision, and writing—original draft.

### Conflict of Interest Statement

The authors declare that they have no conflict of interest.

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
