## [Reviewer comments · Life Science Alliance]

Life Science Alliance

An essential role for α 4A-tubulin in platelet biogenesis

Catherine Strassel, Maria Magiera, Arnaud Dupuis, Morgane Batzenschlager, Agnes Hovasse, Irina Pleines, Paul Guéguen, Anita Eckly, Sylvie moog, Léa Mallo, Quentin Kimmerlin, Stephane Chappaz, Jean-Marc Strub, Natarajan Kathiresan, Henri de la Salle, Alain Van Dorselaer, Claude Ferec, Jean-Yves Py, Christian Gachet, Christine Schaeffer, Benjamin Kile, Carsten Janke, and Francois Lanza

DOI: 10.26508/lsa.201900309

Corresponding author(s): Francois Lanza, EFS Alsace

Review Timeline:

Submission Date:	2019-01-16
Revision Received:	2019-01-16
Editorial Decision:	2019-01-17
Revision Received:	2019-01-23
Accepted:	2019-01-23

Scientific Editor: Andrea Leibfried

Transaction Report:

Please note that the manuscript was previously reviewed at another journal and the reports were taken into account in the decision-making process at Life Science Alliance. Since the original reviews are not subject to Life Science Alliance's transparent review process policy, the reports and author response cannot be published.

January 17, 2019

RE: Life Science Alliance Manuscript #LSA-2019-00309-T

Dr. Francois Lanza
EFS Alsace
INSERM UMR S949
10 rue Spielmann
Strasbourg F-67065
France

Dear Dr. Lanza,

Thank you for submitting your revised manuscript entitled "An essential role for α 4A-tubulin in platelet biogenesis" to Life Science Alliance. Your work was previously reviewed twice at a different journal and you've revised your work in response to the criticisms raised by those reviewers.

The reviewers who evaluated your manuscript elsewhere thought that a link to human disease was lacking. They furthermore noted discrepancies between defective proplatelet formation in the in vitro assays and platelet count in vivo and they requested addition of some controls. You have revised the work twice, adding patient data that now link your mouse-based observations to a mild version of human macrothrombocytopenia. While the discrepancies between in vivo and in vitro results couldn't be fully resolved, the controls were added and you clearly stated in the manuscript text open questions. We would thus be happy to publish your paper in Life Science Alliance pending final revisions necessary to meet our formatting guidelines:

- please upload all figure files (also SFigures) as individual files
- please incorporate the supplementary methods into the main manuscript file
- please correct legend for figure 3 ('C' instead of the current 'E' label)
- please add a callout in the text to figure S4
- please link your ORCID ID to your profile in our submission system, all corresponding authors should link their ORCID ID, please. You should have received a message with instructions on how to do so.
- please make sure that there are not discrepancies
- please add author contributions in our submission system for CJ

A. FINAL FILES:

-- High-resolution figure, supplementary figure and video files uploaded as individual files: See our detailed guidelines for preparing your production-ready images, <http://life-science-alliance.org/authorguide>

B. MANUSCRIPT ORGANIZATION AND FORMATTING:

Full guidelines are available on our Instructions for Authors page, <http://life-science-alliance.org/authorguide>

Sincerely,

Andrea Leibfried, PhD
Executive Editor
Life Science Alliance

Meyerhofstr. 1
69117 Heidelberg, Germany
t +49 6221 8891 502
e a.leibfried@life-science-alliance.org
www.life-science-alliance.org

January 23, 2019

RE: Life Science Alliance Manuscript #LSA-2019-00309-TR

Dr. Francois Lanza
EFS Alsace
INSERM UMR S949
10 rue Spielmann
Strasbourg F-67065
France

Dear Dr. Lanza,

Thank you for submitting your Research Article entitled "An essential role for α 4A-tubulin in platelet biogenesis". It is a pleasure to let you know that your manuscript is now accepted for publication in Life Science Alliance. Congratulations on this interesting work.

DISTRIBUTION OF MATERIALS:

Again, congratulations on a very nice paper. I hope you found the review process to be constructive and are pleased with how the manuscript was handled editorially. We look forward to future exciting submissions from your lab.

Sincerely,
